https://doi.org/10.1038/s41467-019-11887-2　　**OPEN**

# Multistep nucleation and growth mechanisms of organic crystals from amorphous solid states

Hongliang Chen [1,5], Mingliang Li[1,5], Zheyu Lu[2,5], Xiaoge Wang[1,5], Junsheng Yang[3], Zhe Wang[2], Fei Zhang[1], Chunhui Gu[1], Weining Zhang[1], Yujie Sun[3], Junliang Sun [1], Wenguang Zhu [2] & Xuefeng Guo [1,4]

Molecular self-assembly into crystallised films or wires on surfaces produces a big family of motifs exhibiting unique optoelectronic properties. However, little attention has been paid to the fundamental mechanism of molecular crystallisation. Here we report a biomimetic design of phosphonate engineered, amphiphilic organic semiconductors capable of self–assembly, which enables us to use real-time in-situ scanning probe microscopy to monitor the growth trajectories of such organic semiconducting films as they nucleate and crystallise from amorphous solid states. The single-crystal film grows through an evolutionary selection approach in a two-dimensional geometry, with five distinct steps: droplet flattening, film coalescence, spinodal decomposition, Ostwald ripening, and self-reorganised layer growth. These sophisticated processes afford ultralong high-density microwire arrays with high mobilities, thus promoting deep understanding of the mechanism as well as offering important insights into the design and development of functional high-performance organic optoelectronic materials and devices through molecular and crystal engineering.

[1] Beijing National Laboratory for Molecular Sciences, State Key Laboratory for Structural Chemistry of Unstable and Stable Species, College of Chemistry and Molecular Engineering, Peking University, Beijing 100871, P. R. China. [2] ICQD, Hefei National Laboratory for Physical Sciences at the Microscale, Key Laboratory of Strongly-Coupled Quantum Matter Physics of Chinese Academy of Sciences, School of Physical Sciences, University of Science and Technology of China, Hefei, Anhui 230026, P. R. China. [3] State Key Laboratory of Membrane Biology, Biomedical Pioneering Innovation Center, School of Life Sciences, Peking University, Beijing 100871, P. R. China. [4] Department of Materials Science and Engineering, College of Engineering, Peking University, Beijing 100871, P. R. China. [5] These authors contributed equally: Hongliang Chen, Mingliang Li, Zheyu Lu, Xiaoge Wang. Correspondence and requests for materials should be addressed to J.S. (email: junliang.sun@pku.edu.cn) or to W.Z. (email: wgzhu@ustc.edu.cn) or to X.G. (email: guoxf@pku.edu.cn)

The natural and spontaneous assembly of amphiphilic molecules (each is composed of a hydrophilic head and a hydrophobic tail) into continuous, precise, complex and functional structures has inspired scientists to study the intrinsic mechanism of biomimetic self-assembly[1–4]. Molecular engineering has enabled synthetic chemists to mimic liquid crystalline structures through the self-assembly of predesigned molecules from their anisotropic amorphous states[5–7], yet little attention has been given to the fundamental mechanism by which they nucleate on the surfaces. To date, the crystallisation mechanism of organic materials largely draws experience from the classical theory of inorganic chemistry[8–10] and is still ill-understood. The larger components in organic and biological materials make the crystal growth process more complex than that of inorganic materials[11–16]. Inspired by the fact that the inherent chemical structure duality and the hydrophobic effect enable phospholipids to self-aggregate into a lamellar bilayer structure, we demonstrate in this study a series of phosphonate engineered, amphiphilic organic semiconductors (OSCs) that can grow via a process termed as crystallisation-driven self-assembly (CDSA)[17–19] to form ultralong, high-density and highly ordered microwire arrays. By integrating this molecular design with real-time in situ atomic force microscopy (AFM) and film X-ray diffraction (XRD), we successfully imaged the entire self-assembly trajectories and the kinetics of the crystallised films at the minute timescale under ambient conditions.

## Results

### Syntheses and characterisations of $C_nP$-BTBT.
Each amphiphilic OSC (benzo[b]benzo[4,5]thieno[2,3-d]thiophene, named $C_nP$-BTBT, $n = 3–11$, Fig. 1a right) consists of a rigid π-backbone and a flexible phosphonate engineered alkyl tail with the superior self-assembly property (syntheses and characterisations, see Supplementary Note 1, Supplementary Figs. 1–5, Supplementary Tables 1–3). This bioinspired design has two considerations: (i) strong π–π interactions between BTBT components facilitates the growth of highly crystalline semiconducting films or crystals from their amorphous states. However, the rigid nature of the π-systems leads to phase transition at high temperatures. This limits the use of AFM, which is generally restricted to ambient conditions and room temperature[11,20]. (ii) The fluidic nature of amphiphilic phosphonate segments induces molecules to exhibit rapid lateral diffusion along the layer via weak noncovalent interactions of hydrophobic tails, but often complicates high-resolution experimental studies of the diffusion process. Therefore, the key to our success in revealing such a CDSA mechanism is the precise balance of the rigidity of the π-systems and the fluidity of the phosphonate segments, making it possible for real-time in situ AFM imaging of the growth trajectories on the surfaces (AFM data processing and code, see Supplementary Note 2, Supplementary Fig. 6). Indeed, through molecular engineering, the phase transition of $C_nP$-BTBT is adjusted to occur in a narrow temperature window (~50–60 °C) in differential scanning calorimetry solid–solid transitions (Supplementary Figs. 4, 5 and Supplementary Table 3). Importantly, these molecules undergo spontaneous phase transition and generate complex micelle architectures with π-conjugated BTBT cores and segmented alkyl phosphonate coronas in a manner that recalls the key feature of living CDSA[17–19]. These micelle architectures in turn serve as mass carriers in long-range migration between separated domains, which is the underlying nonclassical mechanism of crystal evolution. Although the time-dependent evolution seemed to be complicated, the experimental observation of intermediate steps in prenucleation and crystal evolution is an important

implication for the growth of high-quality OSC films and micro/nanostructures.

### Five-step on-surface growth trajectory.
Self-assembly or the nucleation of continuous films is commonly described by the classical theory[20,21] in which a nascent phase (termed nucleus) emerges from the solution in a single step (Fig. 1a left), or through nonclassical pathways[22,23], such as oriented attachment of nanoclusters, followed by diffusion-controlled surface reconstruction (Fig. 1b). However, our real-time in situ AFM experiments showed an unambiguous five-step crystal growth trajectory, bridging sequential classical and nonclassical mechanisms (Fig. 1c). Figure 1d shows a series of six AFM frames collected at different times during the $C_7P$-BTBT assembly, where their shape evolution at the surface is clearly visible. We chose $C_7P$-BTBT as a representative because it displayed the optimal self-assembly properties on the basis of systematic investigations including single-crystal structures (Supplementary Fig. 7, Supplementary Table 4), 2D grazing incidence XRDs (2D-GIXD) (Supplementary Fig. 8, Supplementary Table 5), and film XRD (Supplementary Fig. 9). Between the initial as-spin coated liquid-like droplet (Fig. 1d, 0 h) and the final crystallised films (Fig. 1d, 18.13 h), five sequential steps appeared, which can be categorised into three stages: prenucleation (Stage 1), mass transport (Stage 2) and structural reorganisation and layer growth (Stage 3). In Stage 1, a real-time AFM method was applied to directly observe distinguishable steps in the prenucleation pathway, indicating complex nonclassical pathways. Initially, we detected very small droplets on the surface (Supplementary Fig. 10a–c) that disappeared in the next frame of the same area within several minutes (<10 min). The fast disappearance of these species occurred due to droplet collapsing/flattening into desolventised nanoplates (Fig. 1d, 0.07 h, Supplementary Fig. 10d–f) and their subsequent coalescence into a loose fully covered amorphous base film (Fig. 1d, 0.22 h).

Between Stage 1 and Stage 2, dense thick islands nucleated from the base film (Fig. 1d, 0.22 h; Supplementary Fig. 11). Following this, AFM results confirmed the formation of crystallised islands on the surface in a mass transport process (Stage 2) in two steps: (i) the base film demixed into thick and thin islands, separating away from each other (Fig. 1d, 2.32 h); (ii) thin islands were transported to thick islands and finally vanished (Fig. 1d, 12.08 h). It is remarkable to observe the long-range migration of organic clusters on the surface. This yields an Ostwald ripening approach, which enables the growth of the thick islands at the cost of the thin islands. It is worth mentioning that Ostwald ripening is a thermodynamically driven phenomenon that is generally observed in solution but has now been realised in the 2D solid state[24,25]. This stage will be discussed in detail in the following section. Finally, when the thick island was isolated beyond the diffusion length of small molecules, they underwent self-reorganisation and layer growth (Stage 3). AFM results demonstrated the self-confined layer growth mode and the etching process of high-energy facets for vertical and lateral growth, respectively, into crystallised films or single-crystal microwires (Fig. 1d, 18.13 h).

### Detailed growth mechanism of each step.
To further identify the detailed growth mechanism, we examined the initial steps of film formation and morphology evolution (Fig. 2a–f). When compared with the crystal growth in solutions or heated AFM, our system achieves its threshold observation window for nucleation at room temperature due to the higher molecular diffusion capability of phosphonate anchors, and thereby accelerates the film

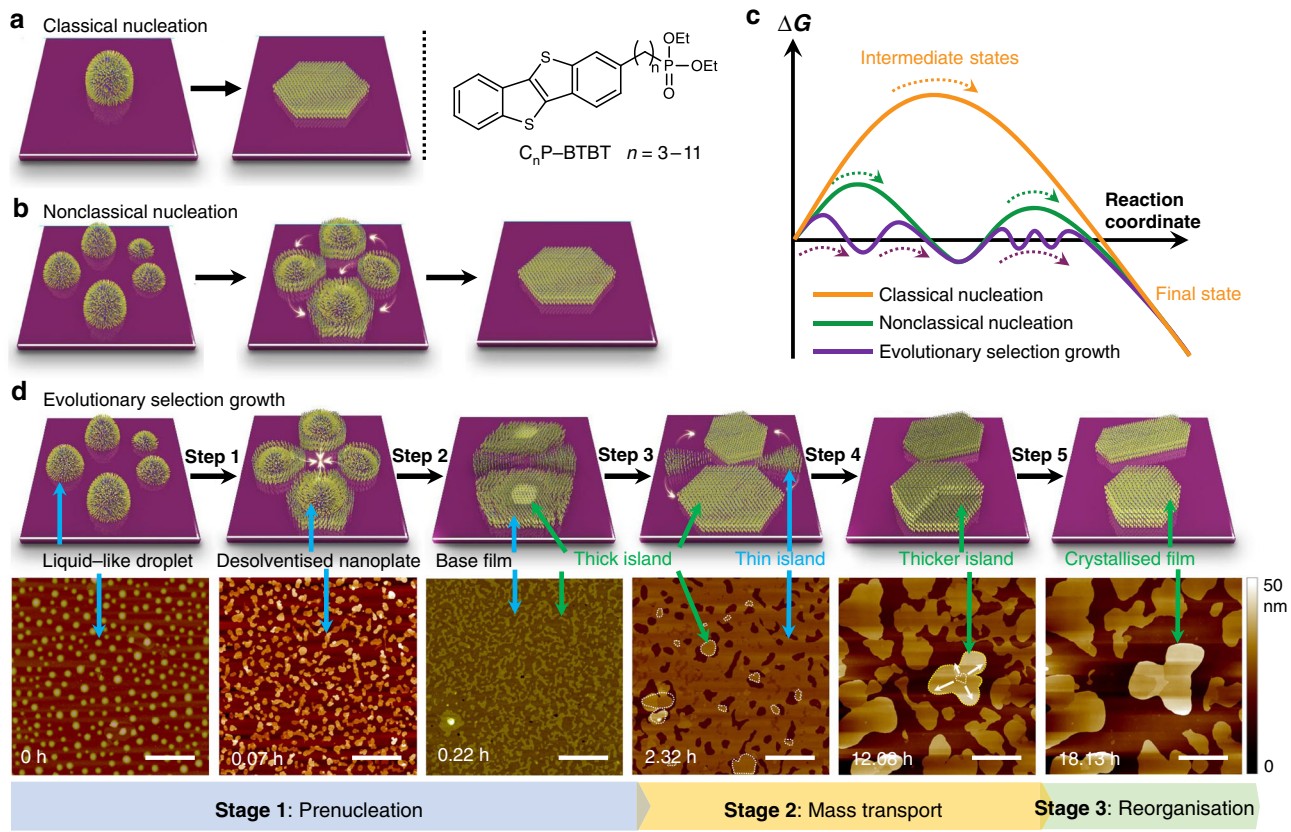

**Fig. 1** Five-step biomimetic self-assembly trajectories of $C_7P$-BTBT on the $SiO_2$ surface. **a** Classical monomer-by-monomer addition (left) and the structure of $C_nP$-BTBT ($n = 3$–11) (right). **b** Nonclassical two-step coalescence mechanism. **c** Plots of the free energy change as a function of reaction coordinate in classical (orange), nonclassical (green) and evolutionary selection (purple) growth models. **d** Evolutionary selection growth approach and time-lapse sequence of representative AFM images showing the morphological evolution of the precursors on the $SiO_2$ surface. From left to right, the images were taken at 0 h (as-spin coated liquid-like droplets), 0.07 h (droplets flattened into pancake-like nanoplates), 0.22 h (nanoplates collapsed into fully covered base film), 2.32 h (base films demixed into thick islands and thin islands), 12.08 h (thin islands shrank while thick islands grew) and 18.13 h (thick islands grew vertically into a crystallised film). Scale bar: 2 μm

formation. This is important because it significantly reduces the lateral drift of AFM. To exclude the tip effects in AFM studies, the setpoint voltage was adjusted to the lowest level to minimise the force applied to the sample while scanning. In addition, for long-term continuous measurements by using AFM, temporal changes in tip geometry must be taken into account. The calibration of AFM tips subjected to 24 h of continuous imaging revealed a less than 2.0% error in height data. We collected sequential AFM images in situ and observed the evolution in morphologies and dimensions. By tracking the areas, thicknesses, and dimensions of the films, we calculated a range of the growth rate ($13.7 \pm 5.0 \, \mu m^2 \, h^{-1}$), which is three orders of magnitude faster than π-conjugated organic thin films under ambient conditions[26] and two order of magnitude slower than the naked lipid bilayer on the surface[27]. This fact well explains the reason why we are able to distinguish all the steps of the nucleation process at our experimental time scale.

Figure 2a shows the prenucleation process when $C_7P$-BTBT solution in chloroform was cast on a flat $SiO_2$ surface (for similar results on a quartz surface, see Supplementary Figs. 12 and 13). In order to observe the whole flattening process and extract the kinetics, we prepared a low–concentration solution (0.5 mg/mL), preventing the rapid coalescence of the films on the basis of systematic concentration-dependent experiments (Supplementary Fig. 14). The classical equilibrium shape of liquid-like droplets is shown in Fig. 2a ($t = 00:00$) (Supplementary Figs. 15, 16,

Supplementary Movie 1). Considering the negligible gravity and uniform pressure for a small droplet, its ideal shape is a spherical cap (Supplementary Fig. 10a–c). Driven by the interfacial energy and hydrostatic pressure, the droplets are gradually flattened into a pancake-like structure (Supplementary Fig. 10d–f), initially 20–30 nm high and 1–2 μm wide (Supplementary Fig. 16a), by spreading on the $SiO_2$ surface, increasing the area, and decreasing the population with time (Supplementary Fig. 16b). In practice, this process is more complicated than we envisioned. At the early stage, the number of nanoplates dropped significantly and eventually settled at a constant value (Supplementary Fig. 17). Although some smaller nanoplates were seen to dissolve completely, the decrease in the number of nanoplates was mainly on account of the coalescence events between individual particles (Fig. 2a). We clearly observed the mergence of two adjacent nanoplates on both $SiO_2$/Si wafer (Fig. 2a, Supplementary Fig. 15) and quartz substrate (Supplementary Fig. 12), which indicates a nonclassical nucleation mechanism. In summary, along with droplets flattening, frequent coalescence events between the flattened nanoplates were observed, yielding a fully covered base film.

Between Stage 1 and Stage 2, we borrowed the theory of spinodal decomposition to explain the film rupture process[28,29]. Figure 2b (Supplementary Fig. 18, Supplementary Movie 2, Supplementary Movie 3) shows representative AFM images covering a broad range of growing times from $t = 0$ to $t = 1$ h

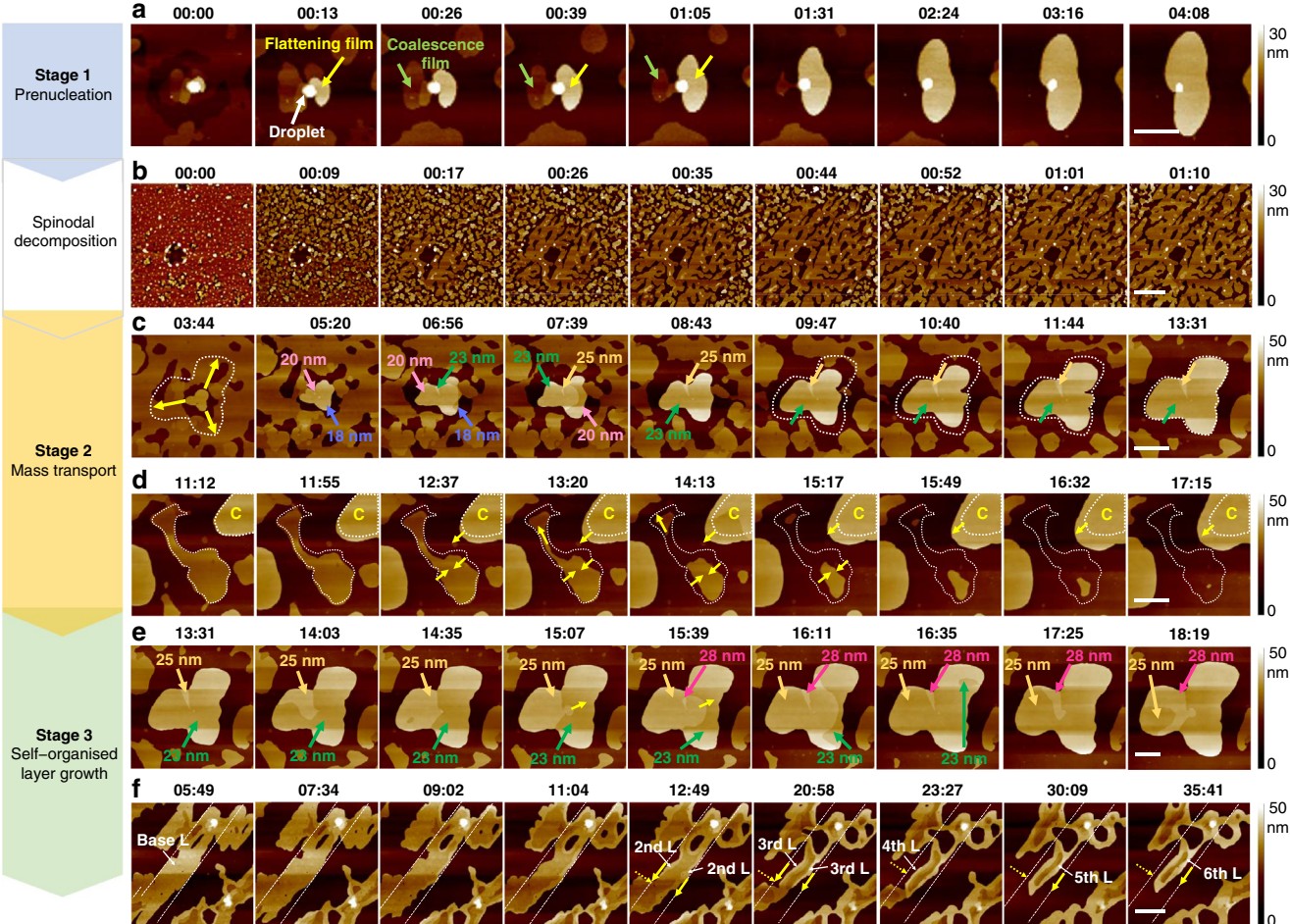

**Fig. 2** Time-resolved film self-growth kinetics at different stages. **a** Flattening of liquid-like droplet into pancake-like films. Scale bar: 2 μm. **b** Spinodal decomposition. Scale bar: 5 μm. **c** Ostwald ripening of a thick film to grow laterally. Scale bar: 2 μm. **d** Disappearance of a thin film. Scale bar: 2 μm. **e** Layer and lateral growth occurring simultaneously. Scale bar: 2 μm. **f** Self-organised layer growth: lateral self-etching for layer growth. Scale bar: 2.5 μm. (35:41 means growth time $t = 35$ h 41 min)

10 min. The bicontinuous surface pattern for the early dewetting stage appears superficially similar to spinodal decomposition patterns found in phase separation measurements and simulations. The film eventually demixed into meta-stable thin films and stable thick films, along with the satellite hole formation. A capillary wave model has been built for investigating thin-film dewetting on solid substrates (Supplementary Figs. 18 and 19).

In Stage 2 (Fig. 2c, d), we found that during prolonged growths, the film shape evolved significantly via a self-selection process, in which the fast-growing thick domains (Fig. 2c) eventually overwhelmed the slow-growing thin domains (Fig. 2d). It resembled the feature of the Ostwald ripening process (Supplementary Movie 4). In general, Ostwald ripening causes smaller species within a critical size distribution to dissolve while larger ones grow up, which is well-understood for crystallisation in solution. In contrast, this process has not been well-studied in solid films despite several unique attributes. In terms of the Ostwald's step rule, stepwise crystallisation occurs such that thermodynamically unstable phases often occur first (small size nucleus), followed by the thermodynamically stable step. Interestingly, we found that the Ostwald ripening of solid films is not just size dependent and the generation of dense thick island plays an important role as well. In this case, the stability of a film would not necessarily scale with the film volume, as in conventional Ostwald ripening. Indeed, materials could transfer from large-

volume thin films to small-volume thick ones (Sample 1: Supplementary Figs. 20 and 21; Sample 2: Supplementary Figs. 22 and 23; Sample 3: Supplementary Figs. 24 and 25). In other words, it is through an evolutionary self-selection growth mode that the thick films grow at the cost of thin films, and eventually overwhelms them to yield a new crystallised film.

Finally, in Stage 3, the layer growth mode is different (Fig. 2e, f, Supplementary Figs. 26–31, Supplementary Movie 5). We considered the standard 2D nucleation and growth mechanism, which describes how a new monolayer of materials nucleates as a 2D island and expands to cover an exposed crystal facet[30]. Commonly, it is believed that the nucleation barrier can be strongly reduced on molecularly narrow facets in comparison with that on wide facets. This leads to a kinetic instability in which the growth in the lateral direction of a platelet is highly accelerated while its thickness remains constant. This is consistent with our experimental results during the ripening process at a short time scale. However, at a prolonged growth time scale, we found that the growth in the vertical direction of a thick domain was highly accelerated while its lateral growth remained suppressed or even contracted. In the case of growing a thick domain, if the distance to the thin domain is within the diffusion length ($<\sim$2 μm), it will continually consume the thin region to grow in both size and thickness (Fig. 2e, Supplementary Figs. 26, 27). However, if the growing domain is totally isolated ($\gg$2 μm), it undergoes a self-confined layer growth mode, during

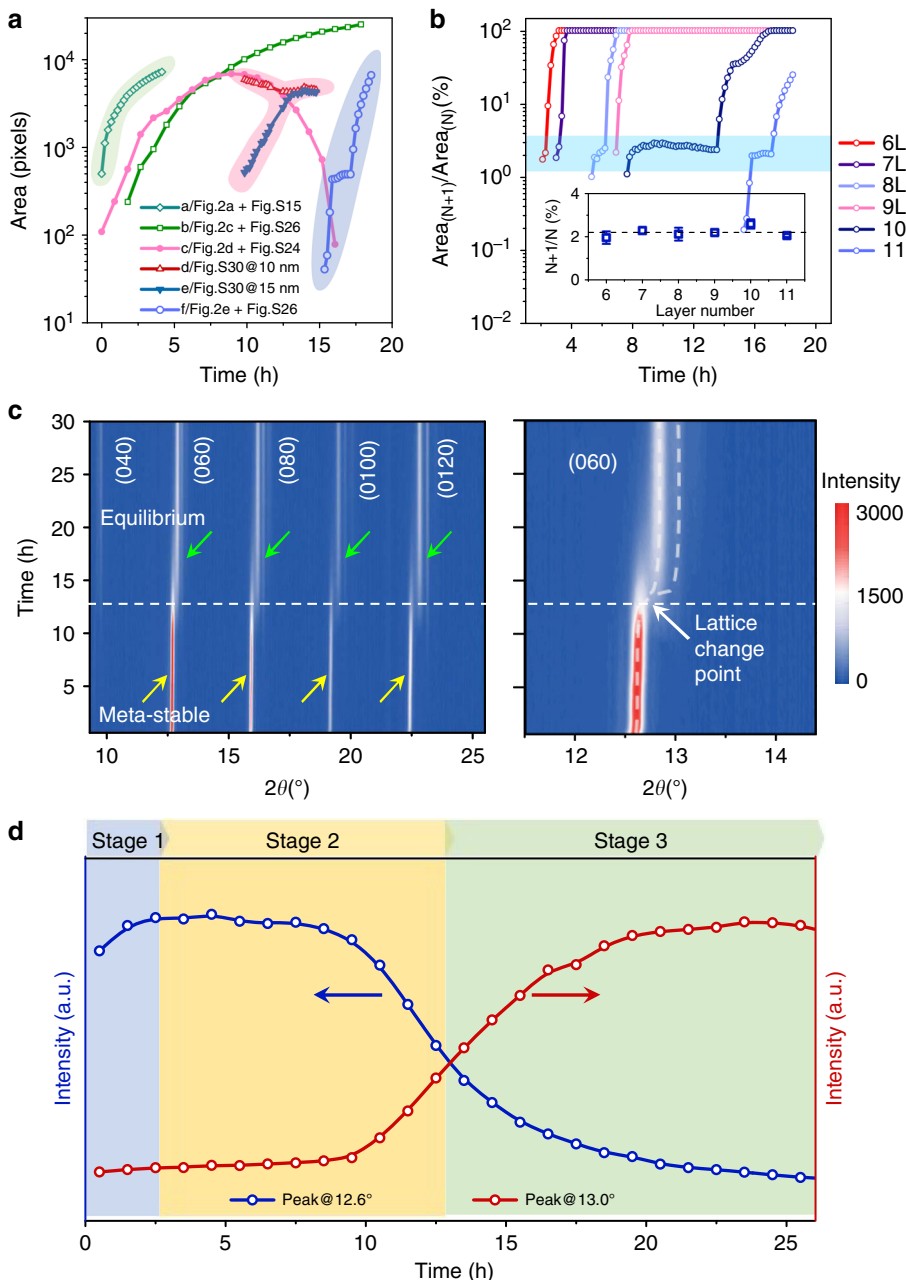

**Fig. 3** Quantitative growth kinetics of the biomimetic self-assembly trajectories. **a** Plots of the domain area as a function of the growth time. These domains are highlighted in Fig. 2a, c, d and 2e; Supplementary Figs. 15, 24, 26 and 30. **b** Area ratio of the upper layer ($N + 1$) and the base layer ($N$) during the layer growth mode. Inset shows a constant area portion of ~2.2%. Error bars represent the standard deviations from an average of three samples. **c** Time evolution of film X-ray diffraction patterns showing the transition of crystalline phases in a 1 mg/mL $C_7P$–BTBT film sample at room temperature (25 °C). Right inset: Magnified lattice evolutions of the decreasing peak at ~12.6° and two newly produced peaks at ~13.0°. Colour scales from blue (low intensity) to red (high intensity). **d** Peak intensity evolutions extracted from the peaks of ~12.6° and ~13.0°, which can be divided into three stages (Stage 1: light blue; Step 2: orange; Step 3: green)

which its high-energy crystal facet is etched and serves as the molecular source to grow the upper layer (Sample 1: Fig. 2f, Supplementary Figs. 28 and 29; Sample 2: Supplementary Figs. 30 and 31; Supplementary Movie 4). This result is consistent with single-crystal results (Supplementary Fig. 7, Supplementary Table 4).

## Discussion

We present the quantitative analyses as follows, focusing on the kinetics of the stepwise growth mechanism. We collected in situ AFM images at discrete time points and estimated the crystallisation kinetics of each step from area statistics of the dynamically changing films (Fig 3a–c, See the Methods section). Figure 3a shows the changes in the domain area as a function of time for the selected domains as examples in which the effective size is reproduced from the pixels in the AFM images. In Stage 1 (the prenucleation process), the fast area evolution kinetics was observed, which includes flattening of liquid-like droplets and fusion of adjacent nanoplates (Fig. 3a, line a; Sample domains in Fig. 2a; full data in Supplementary Figs. 15 and 16). In Stage 2, a thick island evolving by means of Ostwald ripening showed a

continuous increase in size until it reached a saturation stage (Fig. 3a, line b; Sample domains in Fig. 2c; full data in Supplementary Figs. 26 and 27). However, the thin island showed a downward parabolic relationship of the domain size (Fig. 3a, line c; sample domains in Fig. 2d; full data in Supplementary Figs. 24 and 25). This suggests that these regions underwent a size increase in the early stage while scarifying itself in the late stage through Ostwald ripening. We listed other examples (Sample 1 on SiO2 substrate: Supplementary Figs. 20 and 21; Sample 2 on quartz substrate: Supplementary Figs. 22 and 23) that show the self-selection process of fast-growing thick domains which eventually swallows up the slow-growing thin domains and produces a continuous 2D single-crystal film. The fact that Ostwald ripening events are commonly observed among thin domains is attributed to their higher energy because of the larger surface-to-volume ratio and greater mobility of $C_nP$-BTBT on the SiO2 surface.

The case of the layer growth is complicated because both lateral expansion and vertical continuation can occur. The growth kinetics of individual layers is shown in Fig. 3b and Supplementary Fig. 27c. Interestingly, after the nucleus generation, we identified a period of relaxation during which new upper layers ceased to grow (Fig. 3b, light blue box). By extracting the area percentage between the new upper layer and the base bottom layer ($Area_{N+1}/Area_N$), we found a critical area proportion of ~2.2% (Fig. 3b inset). After this relaxation period, the anticlockwise growth of the upper layer resumed until it fully covered the base layer.

To further confirm the growth trajectory, we performed real-time in situ film XRD experiments to monitor the crystal lattice changes at room temperature (25 °C). The experiments were performed in a 1 mg/mL $C_7P$-BTBT film identical to the AFM sample. Each XRD pattern was recorded in the $2\theta$ region of 5–40° for every 30 min at 0.5°/min of scan rate with 50 s of retention interval between the measurements. The peaks corresponding to (040), (060), (080), (0100) and (0120) planes exhibited distinct intensity changes (Fig. 3c). The intensities of the original peaks (yellow arrows) decreased gradually whilst two new Bragg peaks (green arrows) arose at higher angles beside every original peak. Moreover, a remarkable crystal lattice transition point at ~12.5 h was observed which could correspond to the period of self-reorganisation (layer growth) of the isolated islands. After that, the lower $2\theta$ peaks disappeared whilst the higher $2\theta$ peaks became stronger with increasing the time (Fig. 3c right inset, Supplementary Fig. 9). This is a strong evidence that the lower $2\theta$ is from a meta-stable phase since it decreased in intensity and finally disappeared during the growth, while the higher $2\theta$ is strong evidence for a new polymorph. In addition, the clear ($0k0$) Bragg peaks shift in position from $2\theta$ ~12.6° (meta-stable) to $2\theta$ ~13.0° (equilibrium), indicating a smaller intermolecular spacing along (010) direction in the equilibrium state[31]. This is beneficial for OSCs to give high hole carrier mobilities[32].

By extracting the (060) peak intensity evolutions at ~12.6° and ~13.0° (Fig. 3c right inset; Other peaks, see Supplementary Fig. 9), the growth trajectory could be divided into three stages (Fig. 3d), in good agreement with the AFM results (Fig. 1d). Peak intensities at ~12.6° increased from 0 to ~2.5 h, which corresponded to the prenucleation process (Stage 1). However, limited by the time resolution of film XRD, we could not distinguish the flattening (Step 1) and coalescence (Step 2) process. After that, peak intensities kept stable from ~2.5 to ~10 h while decreased gradually from ~10 to ~12.5 h. This time range corresponded to the Ostwald ripening process (Stage 2). Intensity changes of peaks revealed that in the early stage of ripening (2.5–10 h), the crystal lattice remained unchanged in meta-stable state. However, in late stage (10–12.5 h), ripening and layer growth proceeded simultaneously, driving the crystal to an equilibrium stable state.

Eventually, from ~12.5 to 25 h (Stage 3), the peak at ~12.6° disappeared, indicating that the isolated islands underwent a self-confined layer growth to form a dense crystallised film (Stage 3). Both real-time in situ AFM and XRD data prove that the film morphology changed with the crystal lattice on the same time scale throughout the nucleation and crystallisation process.

In order to explain how the thick film formed and broke down, we performed kinetic Monte Carlo (KMC) simulations to investigate the crystal growth based on a simple square lattice model (Supplementary Figs. 32–34, Supplementary Note 3, Supplementary Movie 6). The activation barrier of a molecular diffusion process was determined by the local environment and quantitatively estimated according to the bond counting rule. From the evolution of the total number of islands (Fig. 4a, b), we can see that there are two main periods in the evolution, which we call the two-step Ostwald ripening process. Initially, the molecules formed islands everywhere, resulting in the fast increase in the total number of islands. After reaching a peak, some islands began to merge together and form larger islands. We did observe three different kinds of island evolution routes during the whole process (Fig. 4c). Some were decreasing all the time, some were increasing all the time, and the others were increasing first and then decreasing (Fig. 4c). This result is in good agreement with the experimental observation in Fig. 4d (relevant AFM images, see Supplementary Figs. 24 and 25).

The most useful parameter to characterise the morphology and evolution of islands is the island percentage ($\eta_i$), defined as in Eq. (1)

$$\eta_i \equiv \frac{N_s}{N_a}, \tag{1}$$

where $N_s$ is the number of molecules which belong to an island with the size over a critical value and $N_a$ is the total number of molecules. From our simulations, we found that when the island percentage was used to characterise the evolution of the islands, the terrace barrier has a monotonous effect (Supplementary Fig. 32) while the detachment barrier has a volcano-like influence (Supplementary Fig. 33). The collaborative effect of such two kinetic processes on the island evolution is more complicated as illustrated in Supplementary Fig. 34.

To detect the nanoscopic mass transport process, we performed high-resolution in situ AFM studies between two adjacent thick and thin regions (Fig. 4e). Prior to that, we excluded the effect of residual solvent. Residual chloroform in films can be easily pumped out under ultrahigh vacuum. Three different wafers, which were kept for 24 h in air, a glove box, and ultrahigh vacuum, showed almost the same morphology evolution (Supplementary Fig. 35), proving that the solvent plays a negligible role in the process. In addition, X-ray photoelectron spectroscopy measurements on as-prepared thin films confirm the absence of Cl, which is a fingerprint of chloroform (Supplementary Fig. 36). Figs. 4f–m show sequential AFM images collected under high scanning rates, depicting obvious nonclassical migration trajectories of $C_7P$-BTBT nanoclusters. We directly observed time-dependent movements of spherical nanoclusters, such as those marked with green, which constantly shifted from one location to another on the substrate while their shapes remained unchanged. In addition, we clearly obtained the nanocluster peeled off from the edge (marked with red arrow) and moved toward up right (dash white arrow). Additional important information captured in the AFM images is highlighted Supplementary Fig. 37, where another spherical micelle structure was observed to move on the substrate at the same speed. By extracting the diffusion length and time, we calculated a range of the migration rate (12.5 ± 2.0 nm/min) which clearly

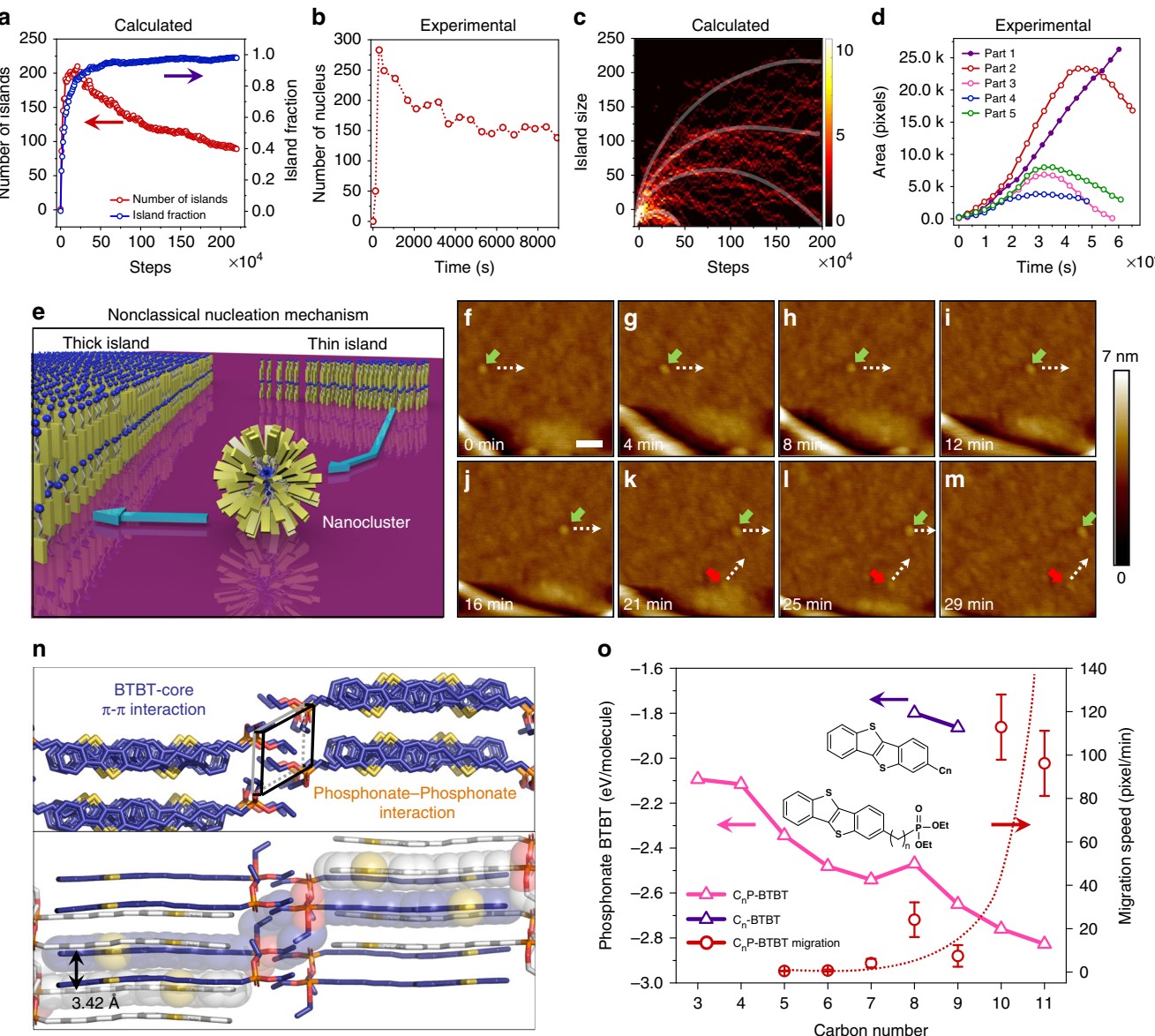

**Fig. 4** Experimental evolution of C₇P-BTBT islands in comparison with kinetic Monte Carlo simulations. **a**, **b** Comparison between simulated (**a**) and experimental (**b**) time-dependent number evolutions of total islands. **c**, **d** Comparison between simulated (**c**) and experimental (**d**) time-dependent evolutions of the island area. **e** Schematic illustration showing the nanocluster-involved nonclassical nucleation mechanism. **f–m** High-resolution AFM images showing evidence of the spherical molecular cluster as mass transport carriers on the surface (green and red arrows marked two nanoclusters; white dash arrows marked the moving directions). Scale bar: 50 nm. **n** Single-crystal structure of C₇P-BTBT showing the BTBT-core π–π interaction and phosphonate–phosphonate coupling (top), and the compressed π–π stacking distance of 3.42 Å (bottom). **o** Binding energy calculation results for CₙP-BTBT (n = 3–11) (pink triangle), Cₙ-BTBT (n = 8 and 9) (blue triangle) and corresponding growth rates of CₙP-BTBT (n = 5–11) (red circle). Error bars are standard deviations of migration speed measured from multiple samples

demonstrated the self-assembly behaviour of the amphiphilic semiconductor and the long-range migration of nanoclusters on the SiO₂ substrate during crystallisation.

Ostwald ripening induced by the migration of adsorbed molecules, via sequential random and short–range (~5 Å) diffusion/hopping across the surface, is a well-known phenomenon in surface science. In our system, however, the long-range migration of nanoclusters cannot be explained by just molecular diffusion because the nanoclusters move as a whole and basically maintain their original shape (rather than gradually disintegrate). Herein, we propose a mechanism for the formation of a nanoscopic mass carrier as illustrated in Fig. 4e. When the meta-stable thin film peels off from the edges, amphiphilic C₇P-BTBT

molecules, which have a hydrophilic phosphonate head, a flexible hydrophobic alkyl chain linker, and a rigid aromatic BTBT tail, reorganise in a 3D manner. The crystallisation of the BTBT backbone leads to the preferred formation of spherical structures via collective effects of phosphonate–phosphonate interaction, hydrophobic interaction of alkyl chains and π–π stacking of BTBT units (Fig. 4n). This biomimetic design simulates the self-assembly behaviours of phospholipid bilayers, which thermodynamically favour the formation of spherical liposomes/micelles structures.

The mechanisms underlying mass transport and nucleation are largely affected by the molecular structure. We have calculated the binding energy of CₙP-BTBT and compared the results with

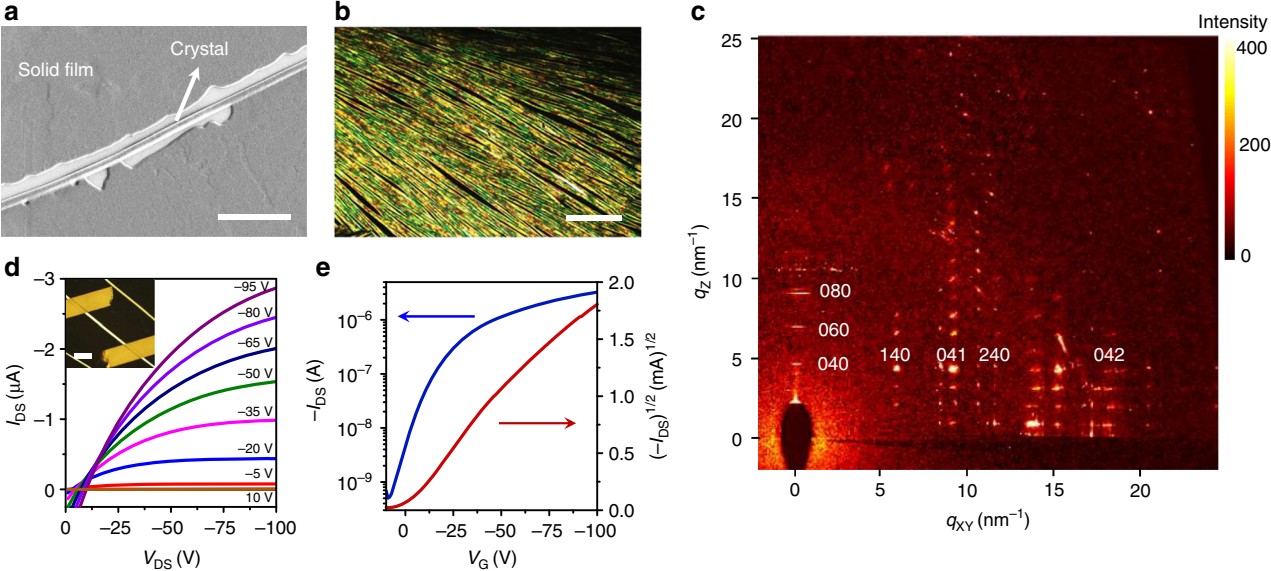

**Fig. 5** Formation of single-crystal microwires from solid films. **a** SEM image showing that an ultralong organic microwire crystallises from solid films with the prolonged growth time. Scale bar: 5 μm. **b** Optical image showing a high-density oriented microwire array. Scale bar: 400 μm. **c** Synchrotron radiation grazing incident X-ray diffraction of single-crystal film/microwire hybrids. More details about unit cell parameters can be found in the Supplementary Table 5. **d** Output characteristics of a microwire single-crystal OFET. Inset shows the polarised optical image of the device. $L = 30$ μm and $W = 1.5$ μm. Gate voltage ($V_{GS}$) ranged from 10 to −95 V in −15 V steps. Scale bar: 10 μm. **e** Transfer characteristics with the source-drain voltage ($V_{DS}$) of −50 V

those of alkyl chain substituted $C_n$-BTBT (Fig. 4o). The absolute values for $C_8$-BTBT and $C_9$-BTBT (Fig. 4o, blue triangle) were much smaller than those of $C_nP$-BTBT (Fig. 4o, pink triangle), implying that the aggregation of $C_n$-BTBT molecules in their films is more difficult. This is consistent with the fact that we hardly observed the mass nucleation of naked $C_n$-BTBTs under ambient conditions. In the $C_nP$-BTBT system, we found an increasing trend in binding energy (Fig. 4o, pink triangle) and a consistent increasing trend in growth rate (Fig. 4o, red circle, Supplementary Figs. 39–44) upon elongation of the alkyl linkers. However, an exception occurred at $n = 8$, probably because of the large torsional angle (~73.59°) between the BTBT backbone and the octyl phosphonate substituent (Supplementary Fig. 38). This is an interesting phenomenon resembling the homeoviscous adaptation in cell biology, which is a fundamental biophysical determinant of membrane fluidity balanced between saturated and unsaturated fatty acids.

After mechanism understanding of solid-state crystallisation, by utilising high-concentration samples for the growth over a long time, we clearly observed the formation of single-crystal microwires from solid films as shown in the scanning electron microscopic (SEM) image (Fig. 5a). Figure 5b demonstrates a highly aligned microwire array of $C_7P$-BTBT with the length extending over the whole substrate after spontaneous growth in the glove box for a week (Supplementary Fig. 45). The large-area cross-polarised optical micrographs (OM) of the microwire arrays show uniform colour over the entire area. Detailed morphologies were further evaluated by SEM (Supplementary Fig. 46) and AFM (Supplementary Fig. 47), showing a uniform width of ~1.2 μm with straight and smooth edges. To further examine their crystalline quality, 2D-GIXD was performed (Fig. 5c, Supplementary Fig. 8), which revealed several distinct high-order spots in the direction of $q_z$ (out of plane) at a given $q_{xy}$ (in-plane). The observation of Bragg reflections up to the 14th order indicates a highly crystalline nature, in contrast to detected polycrystalline films that produce diffuse scattering

around Debye rings[33]. These results provide unambiguous evidence that the microwire arrays are single-crystalline with a long-range translational symmetry, implying the good electronic property as described below.

We have fabricated organic field-effect transistors (OFETs) based on highly aligned microwires. Typical output and transfer characteristics for an OFET (channel length/$L$: 30 μm and channel width/$W$: 1.5 μm) are presented (Fig. 5d, e, Supplementary Fig. 48). Surprisingly, the obtained highest hole mobility is ~8.6 cm$^2$ V$^{-1}$ s$^{-1}$ with a high on/off ratio of the order of ~10$^5$ without any further optimisation. A statistical investigation based on over 30 individual devices revealed an average mobility up to ~5.3 cm$^2$ V$^{-1}$ s$^{-1}$ (Supplementary Fig. 48), which is the highest among biomimetic OSCs and is comparable with thermal or solvent annealed BTBT-based semiconducting materials[34]. We expect improved FET properties of these microwire arrays with special attention paid to the device fabrication process.

In summary, this work demonstrates the dynamic on-surface self-assembly of bioinspired solid-state OSCs with nanometre resolution. The nucleation and crystallisation details gleaned from time-resolved in situ AFM/XRD measurements provide unambiguous evidence for a concerted process that bridges two schools of thoughts: a classical mechanism based on the addition of molecules and a nonclassical route involving direct attachment of molecular clusters. The dynamic crystallisation from solids reveals a series of complex events that transcend conventional mechanisms involving 2D layer nucleation and spreading. The mechanisms and molecular engineering model applied here can be naturally extended to a broad class of organic optoelectronic materials and devices.

## Methods
**Materials**. All reagents and chemicals were obtained from commercial sources and used without further purification unless otherwise noted. The synthetic route of $C_nP$-BTBT ($n = 3$–11) is outlined in Supplementary Note 1. All reactions were performed under an inert atmosphere of argon in dry solvents by using standard

Schlenk techniques. The laboratory single-crystal XRD (SXRD) data for samples $C_nP$-BTBT ($n = 3, 4, 5, 6, 9, 10$ and $11$) were collected at 180 K on a Rigaku Oxford XtalAB instrument (Mo K$\alpha$, $\lambda = 0.71073$ Å, graphite monochromator) and the data reduction were performed by CrysAlisPro programme (version 1.171.39.9f). The synchrotron SXRD for $C_nP$-BTBT ($n = 7$ and $8$) samples were collected at Beamline BL17B1 of Shanghai Synchrotron Radiation Facility (SSRF) with $\lambda = 0.6525$ Å at 100 K and the data were processed with HKL3000 software. The structure was solved by direct methods and refined by minimising the sum of squared deviations against $F^2$ using a full-matrix technique with the SHELXL-97 programme.

**Film sample preparation**. $C_nP$-BTBT solution was prepared by dissolving the solids in chloroform at room temperature (25 °C). The solution was filtered through the PTFE membrane with a pore size of 0.45 μm. It was then dropped on to the silicon substrate and spin coated at a speed of 3000 rpm, giving a meta-stable initial state film. For real-time AFM and film XRD experiments, the film should be introduced to the equipment within 1 min. Film XRD data were collected on PANalytical high-resolution PXRD. GIXD data were obtained at beamline BL14B1 of the SSRF at a wavelength of 1.2398 Å.

**AFM measurements**. The in situ real-time AFM experiments were performed at room temperature (25 °C). The morphology of thin films was investigated by a ScanAsyst model AFM (Bruker Dimension Icon with Nanoscope V controller) under ambient conditions. The probe consisted of a sharp silicon tip (SCANA-SYST-AIR; thickness: 2.5–8.0 μm, length: 115 μm, width: 25 μm; resonance frequency: 70 kHz; spring constant: 0.4 N m$^{-1}$) attached to a silicon nitride cantilever. Scan frequencies of 1 Hz for good imaging conditions. A loading force of ~150 pN were applied to exclude the tip effects. AFM images were analysed with the software Nanoscope III. AFM data were analysed further using a custom Matlab code based on marker-based watershed segmentation. Details of the processing procedure and code can be found in Supplementary Note 2.

**Fabrication and measurement of microwire FET**. Top-contact/bottom-gate microwire devices were fabricated in situ $n^+$-Si/SiO$_2$ substrates where $n^+$-Si and SiO$_2$ were used as the gate electrode and gate dielectric, respectively. The gold source and drain contacts (100 nm) were glued on the substrate onto the surface of the microwire. The channel length ($L$) and width ($W$) were 30 and 1.5 μm, respectively. The transistor characteristics were obtained at room temperature in air by using a standard probe station and semiconducting parameter analyser (Agilent 4155C).

**Theoretical calculation**. We used KMC methods to simulate the crystal growth. A $200 \times 200$ square lattice with periodic boundary condition based on the terrace-step-kink model is used. The first-principles calculations of binding energy were performed by using the Vienna Ab Initio Simulation Package, which is based on the density functional theory and plane wave basis sets with the projector-augmented wave method. The exchange and correlation functional was treated by using the Perdew–Burke–Ernzerhof parameterisation of generalised gradient approximation, including the van der Waals corrections as parameterised in the semiempirical DFT-D3 method, for total energy calculations. Details could be found in Supplementary Note 3.

## Data availability

All the data generated or analysed during this study are included in this published article (and its Supplementary Information files) or available from the authors upon reasonable request. The crystallographic data in this study have been deposited in the Cambridge Structural Database under entry IDs CCDC 1943567–1943575.

## Code availability

The complete marker-based watershed segmentation Matlab code used in the current study are available in Supplementary Note 2, which includes full documentation for running the code and reproducing results.

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

## Acknowledgements

We acknowledge primary financial supports from National Key R&D Programme of China (2017YFA0204901, 2017YFA0204904 and 2016YFA0301004), the National Natural Science Foundation of China (Grants 21727806, 21621061, 11674299, 11634011 and 11374273), the Natural Science Foundation of Beijing (Z181100004418003), the Fundamental Research Funds for the Central Universities (Grants WK2340000063, WK2340000082 and WK2060190084) and the interdisciplinary medicine Seed Fund of Peking University. The authors thank beamline BL14B1 (Shanghai Synchrotron Radiation Facility) for providing the beam time.

## Author Contributions

X.G., H.C., W.G.Z. and J.S. conceived and designed the experiments. H.C. and M.L. performed the material synthesis, device fabrication and most of the device characterisations. W.G.Z., Z.L. and Z.W. provided the theoretical results. J.S. and X.W. did the XRD measurements. J.Y. and Y.S. did the fluorescence characterisation. F.Z. did the thermal measurements. H.C. and C.G. did the matlab data analysis. H.C. and W.N.Z. did the GIXD measurements. H.C., M.L., Z.L., X.W., J.S., W.G.Z. and X.G. analysed the data and wrote the paper. All the authors discussed the results and commented on the paper.

## Additional information

**Competing interests:** The authors declare no competing interests.

