## [Peer Review File · Nature Communications]

Reviewers' comments:

Reviewer #1 (Remarks to the Author):

The manuscript reports in a systematic and elaborate investigation of crystallization-driven self-assembly mechanism of organic semiconductors with use of time-resolved AFM movies and real-time XRD. The combination of experiments and modelling is a particular strength of this manuscript. They propose a 5-step crystal growth trajectory which includes three stages: Prenucleation / Mass transport / 3-D Reorganization. Particularly, in Page 15 Figure 4, the authors present the movement of nanoclusters on surface without losing the shape, which is an exciting result and classify the nucleation mechanism as a nonclassical way. Furthermore, they extended their work to the application of single crystal organic field-effect transistors and obtained good electrical performance. The manuscript is very well-organized and clearly written. I believe this manuscript can make an excellent contribution to the field, and is acceptable for publication in Nature Communications after addressing the following issues.

1. In Line 83, the phase transition in DSC occurs at $\sim 50\text{--}60$ oC, can it still be regarded as a "room temperature window"?
2. The contrast of AFM in Figure 1d (0.22 h) is not that obvious. Could there be a more clear image?
3. The time scale of Figure S21b doesn't match the AFM frames in Figure S20. The unit should be "min" instead of "s", perhaps.
4. The formation of single crystal microwires should be specified in detail. The authors mentioned that the wires formed in weeks, it is hard to probe the formation by AFM of course. Perhaps the optical microscopy or SEM results are needed to demonstrate the process.
5. In Line 281, references are needed to explain why "this is beneficial for organic semiconductors".

Reviewer #2 (Remarks to the Author):

Manuscript ID: NCOMMS-19-14371-T

Recommendation: major revision

Comment: In this work, the nucleation and crystallization process of amphiphilic OSC CnP-BTBT under amorphous solid states were investigated by using time-resolved in-situ AFM/XRD measurements. The authors proposed five-step on surface growth trajectory during the self-assembly of solid-state OSCs that explain with sequential classical and nonclassical mechanisms. Afterward, the crystallized microwires with good electronic property were prepared from solid films. The work of this manuscript is very systematic and enormous, providing rich experimental data for the crystallization study of solid-state OSCs. Although the results are beautiful, before being published, the manuscript still needs some modification. The detailed comments are listed as following:

1. The originality of this work should be more emphasized. The manuscript is mainly a description of the experimental results and few novel ideas have been proposed. The specific mechanisms of nucleation are rarely discussed just by virtue of a classical mechanism based on the addition of molecules. The spinodal decomposition and the Ostwald ripening process are used for the explanation of Mass transport process.
2. The effects of the strong n-n interactions and the fluidity of the phosphonate segments on the crystallization or self-assembly are rarely discussed. Please give the differences of these effects at each step.
3. What are the ambient conditions for the crystallization in this manuscript? How does the authors control the crystallization process?
4. The crystallization steps proposed by the authors is different in different regions of the sample. How is each stage accurately distinguished and defined in in situ AFM? The division of

crystallization stages in Figure 3d is intuitive, but it is difficult to distinguish stages 1 and 2 in Figure 3c.

Listed below are the details of our responses to the referees' comments.

Reviewer 1:

The manuscript reports in a systematic and elaborate investigation of crystallization-driven self-assembly mechanism of organic semiconductors with use of time-resolved AFM movies and real-time XRD. The combination of experiments and modelling is a particular strength of this manuscript. They propose a 5-step crystal growth trajectory which includes three stages: Prenucleation / Mass transport / 3-D Reorganization. Particularly, in Page 15 Figure 4, the authors present the movement of nanoclusters on surface without losing the shape, which is an exciting result and classify the nucleation mechanism as a nonclassical way. Furthermore, they extended their work to the application of single crystal organic field-effect transistors and obtained good electrical performance. The manuscript is very well-organized and clearly written. I believe this manuscript can make an excellent contribution to the field, and is acceptable for publication in Nature Communications after addressing the following issues.

Response: We thank this reviewer very much for his/her high evaluation, insightful suggestions on our manuscript and kind recommendation.

Comment 1: 1. In Line 83, the phase transition in DSC occurs at ~50–60 °C, can it still be regarded as a “room temperature window”?

Response: Thanks for the comment! Sorry about the misunderstanding. The experimental condition of DSC is a little bit different from AFM. The former is a test of bulk materials while the latter is a test of thin films on the surface. In general, the materials on the surface have more freedom to move, which leads to a lower phase transition temperature than DSC. Actually, “room temperature window” is an empirical rule we use to screen the molecules: To make the molecules to self-assemble on the surface “at room temperature”, phase transition temperature obtained from DSC should be better located in the range of 40-60 °C. If it is lower than 40 °C, the materials may exhibit an amorphous liquid state on the surface; however, if it is higher than 60 °C, it is very difficult to observe the mass transport of the materials on the surface. Therefore, in experiments, the “room temperature” means the ambient condition when doing the AFM experiments. Actually, we have observed the phase transition behavior of our material at room temperature (25 °C) and under the ambient condition from which we know that the on-surface phase transition temperature locates near room temperature. To clarify this confusion, we have modified the sentence from “a narrow room temperature window” to “a narrow temperature window”.

Comment 2: 2. The contrast of AFM in Figure 1d (0.22 h) is not that obvious. Could there be a clearer image?

Response: Thanks for the kind suggestion! We have adjusted the contrast and color bar of Figure 1d to make it clearer. Please see the change in Figure 1d on Page 6.

Comment 3: The time scale of Figure S21b doesn't match the AFM frames in Figure S20. The unit should be "min" instead of "s", perhaps.

Response: Thanks for the good correction! We have corrected the time unit in Figure S21 from "s" to "min" accordingly. Please see the change on Page S28 in the Supplementary Information.

Comment 4: In Line 281, references are needed to explain why "this is beneficial for organic semiconductors".

Response: Thanks for the kind suggestion! We have added a reference discussing the charge carrier mobility enhancement induced by the lattice strain in organic semiconductors (*Nature*, **2011**, 480, 504). In this paper, Gaurav Giri *et al.* altered the π - π stacking distance of 6,13-bis(triisopropylsilylethynyl) pentacene (TIPS-pentacene) from 3.33 Å to 3.08 Å. The positive charge carrier (hole) mobility in TIPS-pentacene transistors increased from $\sim 0.8 \text{ cm}^2 \text{ V}^{-1} \text{ s}^{-1}$ for unstrained films to a high mobility of $\sim 4.6 \text{ cm}^2 \text{ V}^{-1} \text{ s}^{-1}$ for a strained film, demonstrating that using solution processing to modify molecular packing through lattice strain should aid the development of high-performance, low-cost organic semiconducting devices. Please see the changes in Page 13 and Page 23.

Reviewer 2:

In this work, the nucleation and crystallization process of amphiphilic OSC CnP-BTBT under amorphous solid states were investigated by using time-resolved in-situ AFM/XRD measurements. The authors proposed five-step on surface growth trajectory during the self-assembly of solid-state OSCs that explain with sequential classical and nonclassical mechanisms. Afterward, the crystallized microwires with good electronic property were prepared from solid films. The work of this manuscript is very systematic and enormous, providing rich experimental data for the crystallization study of solid-state OSCs. Although the results are beautiful, before being published, the manuscript still needs some modification.

Response: We thank this reviewer very much for his/her high evaluation, useful suggestions on our manuscript and kind recommendation.

The detailed comments are listed as following:

Comment 1. The originality of this work should be more emphasized. The manuscript is mainly a description of the experimental results and few novel ideas have been proposed. The specific mechanisms of nucleation are rarely discussed just by virtue of a classical mechanism based on the addition of molecules. The spinodal decomposition and the Ostwald ripening process are used for the explanation of Mass transport process.

Response: Thanks for the kind suggestion! Indeed, observing the *in situ* and real-time crystallizing process of organic materials remains a great challenge in experiments. To date, the crystallization mechanism of organic materials largely draws experience from the classical theory of inorganic chemistry and is still ill-understood. The larger components in organic and biological materials make the crystal growth process more complex than that of inorganic materials. Liquid-cell AFM is the most widely used technology to study the nucleation mechanism of organic materials, however, limited to the liquid/solid interface. For air/solid interface which is more general for self-assembly, it is not applicable. We have solved this problem by a biomimetic design of phosphonate engineered, amphiphilic organic semiconductors capable of self-assembly, which enables us to use real-time *in-situ* AFM to monitor the growth trajectories of such organic semiconducting films as they nucleate and crystallize from amorphous solid states in ambient conditions. To clearly show the novelty, we would like to list the creative points as follows:

- (i) We have observed the long-range migration of organic clusters on the surface, which mostly remains limited to the soft materials community and has not yet been explored by solid-state chemists mainly because of the perceived rigidity, slow diffusion, and chemical inertness of solids. However, our results have contributed to dispelling these beliefs to some extent.

- (ii) For the first time, we have demonstrated clearly that organic films crystallized through an evolutionary selection approach with five distinct steps: droplet flattening, film coalescence, spinodal decomposition, Ostwald ripening, and self-reorganized layer growth.
- (iii) Most importantly, high-resolution and high-speed AFM images have provided direct and solid evidence for nonclassical nucleation mechanism at the nanoscale of such organic nanoclusters. This is extremely important for building the nucleation model for organic materials.

Therefore, we believe that the concepts underlying this work will be of interest to a general readership working in the areas of surface sciences, synthetic chemistry, and organic electronics. We have provided the description in the introduction part to emphasize the originality accordingly. Please see the description in Pages 3–4.

In addition, we have more detailed discussions on spinodal decomposition on Pages S26–S27 in the Supplementary Materials. We have borrowed the crystallization theory in solution to describe the crystallization mechanism on the surface.

Comment 2: The effects of the strong π - π interactions and the fluidity of the phosphonate segments on the crystallization or self-assembly are rarely discussed. Please give the differences of these effects at each step.

Response: Thanks for the comment! This is a good question which is related to the fundamental understanding of the nucleation mechanism of organic materials. Listed below are our further explanations as follows.

As a whole, strong π - π interactions between BTBT components facilitates the growth of highly crystalline semiconducting films or crystals from their amorphous states. However, the rigid nature of the π -systems leads to phase transition at high temperatures. This limits the use of AFM, which is generally restricted to ambient conditions and room temperature. On the other hand, the fluidic nature of amphiphilic phosphonate segments induces molecules to exhibit rapid lateral diffusion along with the layer *via* weak noncovalent interactions of hydrophobic tails, but often complicates high-resolution experimental studies of the diffusion process. Therefore, the key to our success in revealing such a nucleation mechanism is the precise balance of the rigidity of the π -systems and the fluidity of the phosphonate segments, making it possible for real-time *in-situ* AFM imaging of the growth trajectories on the surfaces. Indeed, through molecular engineering, we have observed the spontaneous phase transition behavior of our materials at room temperature and under the ambient condition. Importantly, these molecules generate complex micelle architectures with π -conjugated BTBT cores and segmented alkyl phosphonate coronas in a manner that recalls the key feature of living crystallisation-driven self-assembly

(Figure 4e). These micelle architectures in turn serve as mass carriers in long-range migration between separated domains (Figures 4f-4m and Figure S37), which is the underlying nonclassical mechanism of crystal evolution. Finally, on the basis of deep understanding of the mechanism, these sophisticated processes afford ultralong high-density microwire arrays with high mobilities, thus offering important insights into the design and development of functional high-performance organic optoelectronic materials and devices through molecular and crystal engineering.

Figure 4. e, Schematic illustration showing the nanocluster-involved nonclassical nucleation mechanism. f–m, High-resolution AFM images showing evidence of the spherical molecular cluster as mass transport carriers on the surface (Green and red arrows marked two nanoclusters; White dash arrows marked the moving directions).

Fig. S37 | AFM images showing the spherical molecular cluster as mass transport carriers between two domains on surface at room temperature (Green arrows marked four nanoclusters; White dash arrows marked the moving directions; Grey dash lines are the leading lines).

Based on the experimental observations, details of their effects at each step are listed as follows:

- (i) In Stage 1 (Prenucleation process), the fluidity of the phosphonate segments plays an important role. Molecules can form sphere clusters due to the inherent amphiphilic chemical structure. Then these clusters merge into bigger and thicker films in the following coalescence step.
- (ii) In Stage 2 (Mass transport process), the strong π - π interactions and the fluidity of the phosphonate segments work together. The π - π interactions will stabilize the thicker film,

while the fluidity of the phosphonate is the driving force to form spherical nanocluster which serves as the mass carrier.

- (iii) In Stage 3 (Reorganization process), the packing of molecules on the surface is much more like the packing behavior in single crystals. Just as shown in Figure 4n, the π - π interactions and phosphonate-phosphonate interaction will drive the self-assembly of molecules to form long-range ordered films.

Overall, this is a challenging question related to the structure-property relationship in the crystallization theory. Further strong cooperations between experimental and theoretical scientists would significantly promote a better fundamental understanding of the crystallization mechanism.

Comment 3: What are the ambient conditions for the crystallization in this manuscript? How does the authors control the crystallization process?

Response: Thanks for the questions! The ambient condition for the AFM measurement is room temperature (25 °C) in air. There are no heating or cooling processes. Experiments were operated in the atmosphere pressure without inert gas protection.

Understanding the mechanism of crystal growth remains an active area of research in a diversity of fields, including geology, biomineralization, catalysis, and pharmaceuticals (*Science* **2015**, 349, aaa6760; *Annu. Rev. Mater. Res.* **2013**, 43, 359; *Chem. Rev.* **2018**, 118, 3681). Liquid cell electron microscopy is a developing technology that allows the powerful capabilities of the electron microscope to image and analyze the nucleation mechanism of inorganic specimens (*Science* **2015**, 350, aaa9886; *Science* **2014**, 345, 1158; *Science* **2016**, 354, 874; *Science* **2012**, 336, 1014). However, this technique is not applicable for organic materials, because the components of organic materials (molecules) are much larger than that of inorganic materials (atoms) which makes the crystal growth process more complex. In addition, organic materials are more fragile to electron beam illumination. Considering these limitations, atomic force microscope (AFM) was applied as a nondestructive, real-time, and *in situ* technique to image the crystallization mechanism of organic materials (*Science* **2010**, 330, 337; *Nature* **2016**, 536, 446; *Science* **2018**, 362, 1135; *Nat. Chem.* **2019**, 11, 109).

We probed the crystallization process by AFM as follows. The organic film was spin-coated onto the substrate. Then real-time and *in situ* AFM measurements were carried out to probe the spontaneous nucleation and crystallization trajectory. It is the same condition as the film XRD experiment. “Imaging” the crystallization process (*Science* **2009**, 323, 1276; *Science* **2012**, 336, 44; *Science* **2014**, 344, 705) is interesting enough in itself to study the spontaneous mass transport and self-assembly behavior of materials in an isolated system, which is important for

understanding the crystallization mechanism of materials. To precisely regulate the crystallization process, we should know exactly the details of each step, including the qualitative big picture and quantitative crystallization kinetics, which is what we are doing in this work.

Comment 4: The crystallization steps proposed by the authors is different in different regions of the sample. How is each stage accurately distinguished and defined in situ AFM?

Response: Thanks for the comment. The five-step growth mechanism describes the process of the as-spin-coated film evolved into the high crystallized film. It is a “blind scanning” of the sample without choosing or selecting a scan area of the film. AFM is a *microscopic* characterization method. Although we are unable to see the whole picture of the film, we have tried our best to enlarge the scan area to 20×20 μm without losing resolution in AFM experiment (Figure S24). Under this area size, we can find different domains by increasing its area size (Part 1) as well as the domains by decreasing their size (Part 2-5). Then, we focus on the film area evolution of each domain to extract the crystallization kinetics of each step. In fact, *in situ* AFM is very helpful to image or “see” the details of each step. For example, we have seen the coalescence of two nuclei (Figure S12), the dewetting of droplets to form the pancake-like film (Figure S15), the spinodal decomposition process (Figure S18), and even the movement of nanoclusters (Figure 4f-m). By extracting the film area of each domain, we can get the kinetic results of each step. As demonstrated by the huge data in both main text and the Supplementary Materials, the quantitative crystallization kinetics and the big picture of the crystallization mechanism are basically uniform. Furthermore, we have used a real-time film XRD technique to get more information at a *macroscopic* level to accurately distinguish and define each step.

The division of crystallization stages in Figure 3d is intuitive, but it is difficult to distinguish stages 1 and 2 in Figure 3c.

Response: Thanks for the professional comment. This is the resolution limit of our XRD instrument. To get the sufficient diffraction peak intensity on a thin film (1mg/mL sample), the scan rate should be 20-30 min/cycle. However, the prenucleation process in Stages 1 and 2 proceeded very fast and finished in 30 min, during which we can only get one or two intensity data from XRD measurements. Grazing-incidence X-ray diffraction of synchrotron radiation grazing should be helpful to reveal the detailed mechanisms, which we plan to do as the next project.

Finally, we would like to thank all the referees very much for the helpful suggestions, the patience, the time and the kind recommendation.

REVIEWERS' COMMENTS:

Reviewer #1 (Remarks to the Author):

The revisions have fully addressed my questions. I enthusiastically support the publication of the manuscript.

Reviewer #2 (Remarks to the Author):

The authors fully considered my comments in the revised manuscript. So the manuscript is worth of a publication in the journal of Nature Communication.

REVIEWERS' COMMENTS:

Reviewer #1 (Remarks to the Author):

The revisions have fully addressed my questions. I enthusiastically support the publication of the manuscript.

Reviewer #2 (Remarks to the Author):

The authors fully considered my comments in the revised manuscript. So the manuscript is worth of a publication in the journal of Nature Communication.

Response: We would like to thank all the referees very much for the helpful suggestions, the patience, the time, and the kind recommendation